



# 1  Temporary pause in the growth of atmospheric ethane and

# 2  propane in 2015-2018

**Hélène Angot[1,2], Connor Davel[1], Christine Wiedinmyer[3], Gabrielle Pétron[3,4], Jashan**
**Chopra[1], Jacques Hueber[1,5], Brendan Blanchard[1], Ilann Bourgeois[3,6], Isaac Vimont[3],**
**Stephen A. Montzka[4], Ben R. Miller[3,4], James W. Elkins[4], Detlev Helmig[1,5].**
[1]Institute of Arctic and Alpine Research, University of Colorado Boulder, Boulder, CO, USA.
[2]School of Architecture, Civil and Environmental Engineering, École Polytechnique Fédérale de Lausanne, Sion,
Switzerland.
[3]Cooperative Institute for Research in Environmental Sciences, University of Colorado Boulder, Boulder, CO, USA.
[4]NOAA, Global Monitoring Laboratory (GML), Earth System Research Laboratories, Boulder, CO, USA.
[5]Boulder A.I.R. LLC, Boulder, CO, USA.
[6]NOAA, Chemical Sciences Laboratory (CSL), Earth System Research Laboratories, Boulder, CO, USA.
*Correspondence to*: Hélène Angot (helene.angot@epfl.ch)
**Abstract.**
Atmospheric non-methane hydrocarbons (NMHCs) play an important role in the formation of
secondary organic aerosols and ozone. After a multidecade global decline in atmospheric mole
fractions of ethane and propane – the most abundant atmospheric NMHCs – previous work has
shown a reversal of this trend with increasing atmospheric abundances from 2009 to 2015 in the
Northern Hemisphere. These concentration increases were attributed to the unprecedented growth
in oil and natural gas (O&NG) production in North America. Here, we supplement this trend
analysis building on the long-term (2008-2010; 2012-2020) high-resolution (~ 3-hour) record of
ambient air $C_2$-$C_7$ NMHCs from in-situ measurements at the Greenland Environmental
Observatory at Summit station (GEOSummit, 72.58°N, 38.48°W, 3210 m above sea level). We
confirm previous findings that the ethane mole fraction significantly increased by +69.0 [+47.4,
+73.2; 95 % confidence interval] ppt per year from January 2010 to December 2014. Subsequent
measurements, however, reveal a significant decrease by -58.4 [-64.1, -48.9] ppt per year from
January 2015 to December 2018. A similar reversal is found for propane. The upturn observed
after 2019 suggests, however, that the pause in the growth of atmospheric ethane and propane
might only have been temporary. The analysis of 2012-2019 air mass back-trajectories shows that
this pause in mole fraction increases can neither be attributed to changes in atmospheric transport





nor to changes in regional emissions. Discrete samples collected at other northern-hemisphere
baseline sites under the umbrella of the NOAA cooperative global air sampling network show a
similar decrease in 2015-2018 and suggest a hemispheric pattern. Here, we further discuss the
potential contribution of biomass burning and O&NG emissions, the main sources of ethane and
propane, and we conclude that O&NG activities likely played a role in these recent changes. This
study, however, highlights the crucial need for better constrained emission inventories.

**1. Introduction**
Non-methane hydrocarbons (NMHCs) are emitted to the atmosphere by a variety of biogenic and
anthropogenic sources. Their atmospheric oxidation contributes to the production of surface ozone
and aerosols, with impacts on air quality and climate forcing (Houweling et al., 1998). The
abundance of the most abundant atmospheric NMHCs (ethane, propane, i-butane, n-butane, i-
pentane, n-pentane) increased steadily after 1950 until reduced emissions from oil and natural gas
(O&NG) production and emission regulations from diverse sources (*e.g.*, automobiles and
industrial processes) began to be implemented in the 1970s (Helmig et al., 2014). Emission
reductions led to a gradual decline (3-12 % per year) of NMHCs at urban and semi-rural sites in
the last five decades (e.g., von Schneidemesser et al., 2010; Warneke et al., 2012). Accounting for
an approximate atmospheric lifetime (at OH = $6.5 \times 10^5$ molecules/cm$^3$) ranging from 4.5 days
for pentanes to 2 months for ethane, these emission reductions are also reflected in observations
of background air composition, as seen in Northern Hemisphere firn air records (Aydin et al., 2011;
Worton et al., 2012; Helmig et al., 2014): light alkanes increased steadily post 1950, peaking ~50
% above 1950 levels around 1970-1985, and then steadily declined until 2010 to levels that were
close to 1950 levels. After some 40 years of steadily declining atmospheric ethane and propane
mixing ratios, Helmig et al. (2016) reported a reversal in this behavior: the analysis of weekly
discrete air samples has shown that between mid-2009 and mid-2014, ethane abundance at surface
sites in the Northern Hemisphere increased at a rate of 2.9-4.7 % per year. These observations and
conclusions were further substantiated by solar Fourier transform infrared (FTIR) ethane column
retrievals showing similar increases in the mid to upper tropospheric ethane column (Franco et al.,
2015, 2016; Hausmann et al., 2016). The largest increase rates for ethane and propane mixing
ratios were found at sites located in the Eastern United States (U.S.) and in the Northern Atlantic





Region, indicating larger emissions from the central to eastern parts of the U.S., with the likely
sources being increased emissions from shale O&NG extraction operations.
Interestingly, there is a strong latitudinal gradient of absolute NMHC dry air mole fractions – with
highest abundances in the Arctic where atmospheric removal rates are low during the polar winter
(Helmig et al., 2016, 2009; Rudolph, 1995). Despite the sensitivity of the Arctic to pollution
transport from lower latitudes, climate change, and already recognized and further anticipated
feedbacks on the global climate, long-term in-situ atmospheric composition observations within
the Arctic are sparse. A large part of our current knowledge of polar atmospheric chemistry stems
from research aircraft missions and campaign-type observations (e.g., Hartery et al., 2018; Jacob
et al., 2010; Law et al., 2014). However, long-term continuous measurements or regularly repeated
observations with consistent methodology and instrumentation are indispensable for establishing
a baseline record of environmental conditions at clean remote sites and for observing their changes
over time. Such data also serve as a legacy for future research that will rely on comparison with
archived observations of environmental conditions.
In that context, the National Oceanic and Atmospheric Administration (NOAA) Global
Monitoring Laboratory (GML) initiated a cooperative air-sampling network at Niwot Ridge,
Colorado, in 1967 (hereafter referred to as the NOAA/GML Carbon Cycle Greenhouse Gases
(CCGG) network (https://www.esrl.noaa.gov/gmd/ccgg/)). This network is nowadays an
international effort and discrete air samples are collected approximately weekly from a globally
distributed network of sites, including four Arctic sites: Utqiagvik (formerly known as Barrow,
Alaska, USA), Alert (Nunavut, Canada), Summit (Greenland), and Ny-Ålesund (Svalbard,
Norway). These samples are analyzed for $CO_2$, $CH_4$, CO, $H_2$, $N_2O$, and $SF_6$ at GML (e.g., Geller
et al., 1997; Komhyr et al., 1985; Steele, 1991), at the University of Colorado Institute for Arctic
and Alpine Research (INSTAAR) for stable isotopes of $CO_2$ and $CH_4$ (Miller et al., 2002; Trolier
et al., 1996), and, since 2004, for a variety of volatile organic compounds (VOCs) including $C_2$-
$C_7$ NMHCs (Pollmann et al., 2008; Schultz et al., 2015). Since 2014, measurements of ethane and
propane were added to discrete air samples collected under the umbrella of the NOAA/GML
Halocarbons and other Atmospheric Trace Species (HATS) network since 2004
(https://www.esrl.noaa.gov/gmd/hats/flask/flasks.html).
The discrete, typically weekly, air sampling by cooperative global networks have been at the
forefront of studies to identify and quantify long-term trends in the background air abundances of



important trace gases (e.g., Masarie and Tans, 1995; Montzka et al., 2018; Nisbet et al., 2014,
2019). In parallel, higher temporal-resolution in-situ measurements allow the investigation of
source regions and of shorter-term trends at specific sites. Here, we report in-situ 2 to 4-hourly
ambient air $C_2$-$C_7$ NMHCs dry air mole fractions from measurements at the Greenland
Environmental Observatory at Summit station (GEOSummit) by gas chromatography (GC) and
flame ionization detection (FID). Despite the advent of new methods based on optical
measurement (e.g., FTIR spectroscopy) and mass spectrometry (e.g., Photon-Transfer Mass
Spectrometry), GC-FID remains the dominant method in routine VOC observations due to its
stable long-term response characteristics and relatively low maintenance cost (Schultz et al., 2015).
NMHCs were first monitored with high temporal frequency at GEOSummit from 2008 to 2010
with support from the NASA Research Opportunities in Space and Earth Sciences (ROSES)
program (Kramer et al., 2015). NMHC monitoring resumed in 2012 as part of the National Science
Foundation (NSF) Arctic Observing Network program and has been continuous and uninterrupted
until March 2020, providing one of the few high-temporal resolution long-term records of NMHCs
in the Arctic. In this paper, we investigate and discuss seasonal variations, rates of change, and
potential sources of NMHCs in the high Arctic. We also analyze multiyear trace gas data from
other background sites under the umbrella of the NOAA/GML CCGG and HATS sampling
networks to support our findings.

## 2. Materials and Methods

GEOSummit (72.58°N, 38.48°W, 3210 m above sea level) is a research facility located on the
Greenland ice sheet funded by the U.S. NSF and operated in collaboration with the Government
of Greenland (see Fig. 1). The station hosts a diverse array of Geoscience and Astrophysics
research projects (https://www.geosummit.org/instruments) and is the only high altitude remote
atmospheric observatory in the Arctic. Ambient outside air is monitored at the Temporary
Atmospheric Watch Observatory (TAWO), *i.e.,* ~ 1 km south of the research camp.

### 2.1 In-situ NMHC measurements

$C_2$-$C_7$ NMHCs (ethane, propane, iso-butane, n-butane, acetylene, iso-pentane, n-pentane, n-
hexane, benzene, toluene) were analyzed from July 2008 to July 2010 and from May 2012 to
March 2020 by GC-FID using a fully automated and remotely controlled custom-built system.
Ambient air was continuously sampled from a 10 m high inlet on the meteorological tower adjacent



to the TAWO building through a heated (~30°C) sampling line. The sampling frequency increased
from 6 ambient NMHC runs to 12 daily runs in 2018. The GC-FID system, tailored towards the
remote, unattended and long-term operation, is a further development of the instrument described
in detail by Tanner et al. (2006) and Kramer et al. (2015). The instrument relies on a cryogen-free
sample enrichment and injection system. Air was pulled from the tower inlet, and aliquots of the
sample stream were first passed through a water trap (u-shaped stainless-steel treated Silcosteel™
tube cooled using thermoelectric coolers) to dry the sample to a dew point of -20°C, and NMHCs
were then concentrated on a Peltier-cooled (-35°C) multi-stage adsorbent trap. Analysis was
accomplished by thermal desorption and injection onto an $Al_2O_3$ PLOT column for cryogen-free
separation on an SRI Model 8610 GC-FID. Our monitoring effort followed the World
Meteorological Organization (WMO) Global Atmospheric Watch (GAW) quality control
guidelines: blanks and calibration standards were injected every other day from the manifold and
processed in the exact same way as ambient samples. The limit of detection was ~2 ppt (pmol/mol
by volume) for all compounds and no significant blank contamination was ever noticed.
Quantification was based on monthly FID response factors (Scanlon and Willis, 1985) calculated
from the repeated analysis of two independently prepared and cross-referenced standards in use at
any given time. Tables S1 and S2 summarize these response factors along with the associated
relative standard deviation (< 5 % on average for all compounds) for 2008-2010 and 2012-2020,
respectively. The in-situ GC-FID system provided a stable response from 2008 to 2020, with
monthly response factors varying by ≤ 5 % for ethane, propane, and butanes, and by ≤ 20 % for
other compounds over this period. The monitoring program was audited by the World Calibration
Center for Volatile Organic Compounds at the site in July 2017 (https://www.imk-ifu.kit.edu/wcc-
voc/). All reported VOCs results were found to be within the Global Atmospheric Watch program
quality objectives (WMO, 2007).
**2.2 Discrete measurements**
We use here NMHC data from Alert, Utqiagvik, Mace Head (Ireland), Park Falls (Wisconsin,
USA), and Cape Kumukahi (Hawaii, USA; see Fig. 1) collected as part of the NOAA/GML CCGG
(October 2004 to August 2016) and HATS (August 2014 to March 2020) sampling and
measurement programs. Note that we combine here measurements from the two networks.
2.2.1   CCGG discrete sampling and analysis





As described by Steele et al. (1987) and Dlugokencky et al. (1994), air samples are collected
~weekly in pairs in 2.5 L borosilicate flasks with two glass-piston stopcocks sealed with Teflon
O-rings. Flasks are flushed in series for 5 to 10 minutes then pressurized to ~1.2 atm with a portable
sampling system. Samples collected from October 2004 to August 2016 were analyzed at
INSTAAR in Boulder, Colorado, by GC-FID. The analysis, on a HP-5890 series II gas
chromatograph, first involved drying of approximately 600 cc of sample gas by running the sample
gas through a 6.4 mm o.d. stainless steel tube cooled to -25°C. The analytes were then
preconcentrated at -35°C on an adsorbent bed (Carboxen 1000/1016). Samples were thermally
desorbed at 310°C onto a short capillary guard column before separation on an $Al_2O_3$ PLOT
capillary column (0.53 mm × 60 m). Weekly instrument calibrations were performed using
primary calibration standards acquired from the NOAA Global Monitoring Laboratory, the U.K.
National Physics Laboratory, and the U.S. National Institute of Technology. These standards
scales have been maintained since 2006 by regular inter-comparison of standards from these
sources and propagation of the scale with newly acquired standards. Deviations in the response
factors from these different standards were smaller than 5 %, with results for ethane and propane
typically being equal or having less than 2-3 % deviation. Instrument FID response is linear within
the range of observed ambient concentrations. The INSTAAR NMHC laboratory was audited by
the WMO GAW World Calibration Center for VOCs (WCC-VOC, https://www.imk-
ifu.kit.edu/wcc-voc/) in 2008 and in 2016, and both times all measurement results passed the
WMO data quality criteria (WMO, 2007).
2.2.2   HATS discrete sampling and analysis
At GEOSummit, paired borosilicate glass flasks are also pressurized to ~1 atmosphere
overpressure with ambient air as part of the HATS sampling program. At other NH sites,
electropolished stainless-steel flasks are used. All flasks are analyzed by GC with mass
spectrometry analysis with a preconcentration system similar to Miller et al. (2008) to strip water
vapor and $CO_2$ from the airstream prior to injection of condensates (VOCs, halocarbons, solvents,
and other gases) onto a 0.32 mm i.d. GasPro capillary column. Results are tied to a suite of
standards prepared in-house with gravimetric techniques.
**2.3 Ancillary data**
Continuous monitoring of carbon monoxide (CO) has been ongoing at GEOSummit since May
2019 with a cavity ring-down spectroscopy (CRDS) analyzer (Picarro G-2401). A switching





manifold allows regular sampling of ambient air and calibration gases. Three NOAA GML
standards were integrated into the automated calibration. Low (69.6 ppb) and high (174.6 ppb)
calibration points were performed for ~3 minutes every two days, while an intermediate (117.4
ppb) calibration was carried out in between. Using the last minute of each calibration, the low and
high calibration points were used to determine the linear relationship between the certified
calibration values and the analyzer's reported calibration values. The calibration offset (slope and
intercept) was calculated and used to correct the third intermediate calibration point. The mean
absolute difference between the corrected and certified intermediate calibration paired values was
1.6 ppb, *i.e.,* 1.4 %. The minute-averaged CRDS CO ambient air data were corrected using the
calibration offset. The CRDS has a manufacturer-specified precision at 5 seconds, 5 minutes, and
60 minutes of 15, 1.5, and 1 ppb for CO (G2401 Gas Concentration Analyzer | Picarro, 2020).
We also use ethane, propane, tetrachloroethylene ($C_2Cl_4$), and hydrogen cyanide (HCN) data
collected in the free troposphere during the global-scale airborne Atmospheric Tomography
mission (ATom; https://espo.nasa.gov/atom/content/ATom) onboard the NASA DC-8 aircraft
(Wofsy et al., 2018). Canisters collected with the University of California Irvine Whole Air
Sampler (WAS) were analyzed for more than 50 trace gases, including ethane, propane, and
tetrachloroethylene by GC-FID and GC-mass spectrometric detection (Barletta et al., 2020).
Hydrogen cyanide was measured in situ with the California Institute of Technology Chemical
Ionization Mass Spectrometer (CIT-CIMS; Allen et al., 2019). For the purpose of our analysis, we
filtered out data collected over continents, in the marine boundary layer (altitude < 0.4 km), or
corresponding to stratospheric air (ozone to water vapor ratio > 1 ppb per ppm).

205        **2.4 Curve fitting method and trend analysis**

We used the curve fitting method developed by Thoning et al. (1989) and described in detail at
https://www.esrl.noaa.gov/gmd/ccgg/mbl/crvfit/crvfit.html. Briefly, the data were fitted with a
function consisting of a polynomial and series of harmonics to represent the average long-term
trend and seasonal cycle. Residuals from the function were calculated, transformed into frequency
domain with a fast Fourier transform algorithm, then filtered with two low pass filters. One
eliminates harmonics less than ~1 month. When converted back to time domain and added to the
function, it gives a smoothed curve. The other filter eliminates periods less than ~1 year; when
transformed back to time domain and added to the polynomial, it gives the deseasonalized trend
(hereafter referred to as the trend). The Sen's slope estimate of the trend was calculated using





function TheilSen in R package openair (Carslaw and Ropkins, 2012). Note that the p-values and
all    uncertainties    are    calculated    through    bootstrap    simulations
(https://davidcarslaw.github.io/openair/reference/TheilSen.html).
**2.5 Source apportionment analysis**
In order to identify potential source regions, we performed a Potential Source Contribution
Function (PSCF) analysis using the *trajLevel* function in R package openair (Carslaw and Ropkins,
2012). Based on air-mass back-trajectories (see below) and NMHC residuals (see Section 2.4), the
PSCF calculates the probability that a source is located at latitude $i$ and longitude $j$. PSCF solves:
$$PSCF = {m_{ij}}/{n_{ij}} \qquad \text{Eq.1}$$
where $n_{ij}$ is the number of times that the trajectories passed through the cell $(i, j)$ and $m_{ij}$ the
number of trajectories passing through that cell in which the NMHC residual was greater than a
given threshold (90[th] percentile of the measured results distribution). Note that cells with very few
trajectories passing through them have a weighting factor applied to reduce their effect.
For each NMHC in-situ measurement, HYSPLIT (HYbrid Single Particle Lagrangian Integrated
Trajectory; Draxler and Rolph, 2013) 5-day air-mass back trajectories used in the PSCF analysis
were generated using the Python package *pysplit* (Warner, 2018) and processor *pysplitprocessor*
available    at:    https://github.com/brendano257/pysplit    and
https://github.com/brendano257/pysplitprocessor, respectively. The HYSPLIT Lagrangian
particle dispersion model was run from April 2012 to June 2019 using the National Center for
Environmental Prediction Global Data Assimilation System (NCEP GDAS) $0.5° \times 0.5°$
meteorological inputs available at: ftp://arlftp.arlhq.noaa.gov/pub/archives/gdas0p5. We did not
generate back-trajectories for observations after June 2019 due to the unavailability of the GDAS
$0.5° \times 0.5°$ archive.

**3.  Results and Discussion**
**3.1 Seasonal variation**
The seasonal variation of $C_2$-$C_7$ NMHCs at GEOSummit is displayed in Fig. 2. Summer refers to
June-August, fall to September-November, winter to December-February, and spring to March-
May. NMHCs exhibit a strong and consistent seasonal pattern year after year, with maximum mole
fractions during winter and early spring, and a rapid decline towards summer. Anthropogenic
sources of NMHCs do not vary much seasonally (Pozzer et al., 2010). Therefore, the observed



seasonal cycle is primarily driven by the seasonally changing sink strength by the photochemically
formed OH radical (Goldstein et al., 1995) – the dominant oxidizing agent in the global
troposphere (Levy, 1971; Logan et al., 1981; Thompson, 1992). We found a significant correlation
($R^2 = 0.7$, p-value < 0.001) between the mean seasonal amplitude of individual NMHCs and their
lifetime against oxidation by the OH radical (Fig. S1). During the summer period, mole fractions
of the heavier NMHCs were below or close to the GEOSummit in-situ system detection limit (Fig.
2b). As already noted by Goldstein et al. (1995) and Kramer et al. (2015) based on a limited dataset,
the phase of each NMHC is shifted due to the rate of reaction with OH. Ethane, the lightest and
longest lived of the NMHCs shown in Fig. 2, peaks in February/March with a median of 2110 ppt,
and declines to a minimum of 734 ppt in July. Heavier and shorter-lived NMHCs have lower mole
fractions, peak earlier in the year (January/February), and reach a minimum earlier in summer
(June) due to their faster rate of reaction with OH (Chameides and Cicerone, 1978).
Because changes in NMHC sources and sinks can affect the seasonal cycle amplitude, we
investigated whether there is a trend in the NMHC's amplitude at GEOSummit. We focus here on
ethane and propane, the most abundant hydrocarbons in the remote atmosphere after methane.
Figure 3 shows the amplitude of the ethane and propane seasonal cycles, determined as the relative
difference between the maximum and minimum values from the smooth curve for each annual
cycle (Dlugokencky et al., 1997). The peak-to-minimum relative amplitude ranged from 64 to 71
% for ethane and from 92 to 96 % for propane, and there is no indication of a significant overall
trend in amplitude. This range of amplitudes is in good agreement with the literature: the typical
seasonal amplitudes for ethane are on the order of 50 % at mid-latitude sites and can increase up
to 80 % at remote sites (Franco et al., 2016; Helmig et al., 2016). Changes in mole fractions are
further investigated and discussed in the following section.
**3.2 Reversal of ethane and propane rates of change at GEOSummit in 2015**
Ethane is released from seepage of fossil carbon deposits, volcanoes, fires, and from human
activities – with O&NG extraction, processing, distribution, and industrial use being the primary
sources (Pozzer et al., 2010). Based on the inventory developed for the Hemispheric Transport of
Air Pollutants, Phase II (HTAP2, Janssens-Maenhout et al., 2015), biogenic emissions from
MEGAN2.1 (Guenther et al., 2012), and fire emissions from FINNv1.5 (Wiedinmyer et al., 2011),
Helmig et al. (2016) estimated that ~4 %, 18 %, and 78 % of global ethane emissions are due to
biogenic, biomass burning, and anthropogenic sources, respectively. Global ethane emission rates



277 decreased by 21 % from 1984 to 2010 likely due to decreased venting and flaring of natural gas in

278 oil producing fields (Simpson et al., 2012). As a consequence, atmospheric ethane background air

279 mixing ratios significantly declined during 1984-2010, by an average of -12.4 ± 1.3 ppt per year

280 in the Northern Hemisphere. However, the analysis of ten years (2004-2014) of NMHC data from

281 air samples collected at NOAA GML remote global sampling sites (including GEOSummit)

282 showed a reversal of the global ethane trend from mid-2009 to mid-2014 (ethane growth rates >

283 50 ppt per year at 32 sites). This trend reversal was attributed to increased U.S. O&NG production

284 (Helmig et al., 2016). Figure 4a shows the July 2008-March 2020 ethane trend at GEOSummit, as

285 inferred from our in-situ measurements. Note that the same time-series but also showing individual

286 data points can be found in Fig. S2. Ethane mixing ratios at GEOSummit significantly (p-value <

287 0.001) increased by +69.0 [+47.4, +73.2; 95 % confidence interval] ppt per year from January

288 2010 to December 2014. A reversal is, however, evident after 2015: ethane mixing ratios

289 significantly (p-value < 0.001) decreased by -58.4 [-64.1, -48.9] ppt per year from January 2015

290 to December 2018. Data collected after 2019, however, suggest that the pause in the growth of

291 atmospheric ethane might only be temporary. We focus hereafter on the year 2015 reversal. Similar

292 to ethane, a reversal is evident late 2014 for propane (see Fig. 4b): mixing ratios significantly (p-

293 value < 0.001) increased by +47.9 [+32.3, +52.3] ppt per year from January 2010 to June 2014,

294 but significantly (p-value < 0.001) decreased at a rate of -70.5 [-76.1, -65.8] ppt per year from July

295 2014 to July 2016. Propane mixing ratios remained fairly stable (+10.2 [+6.6, +14.6] ppt per year;

296 p-value < 0.001) from July 2016 to December 2019. It should be noted that the pause in the growth

297 of atmospheric ethane and propane at GEOSummit in 2015-2018 is confirmed by independent

298 discrete sampling under the umbrella of the NOAA/GML CCGG and HATS networks (see Fig. 4;

299 solid lines). Figure S3 shows the good agreement ($R^2$ = 0.97 for ethane, $R^2$ = 0.99 for propane)

300 between in-situ GC-FID measurements and discrete samples.

301 The temporary pause in the growth of ethane and propane at GEOSummit could either suggest a

302 change in: i) the OH sink strength, ii) atmospheric transport from source regions and/or iii)

303 natural/anthropogenic emissions.

304 The tropospheric abundance of OH is driven by a complex series of chemical reactions involving

305 tropospheric ozone, methane, carbon monoxide, NMHCs, and nitrogen oxides, and by the levels

306 of solar radiation and humidity (Logan et al., 1981; Thompson, 1992). Building on the comparison

307 of modeled and observed methane and methyl chloroform lifetimes, Naik et al. (2013) showed that



OH concentrations changed little from 1850 to 2000. The authors suggested that the increases in
factors that enhance OH (humidity, tropospheric ozone, nitrogen oxide emissions, and UV
radiation) was compensated by increases in OH sinks (methane abundance, carbon monoxide and
NMHC emissions). More recently, Naus et al. (2020) used a 3D-model inversion of methyl
chloroform to constrain the atmospheric oxidative capacity – largely determined by variations in
OH – for the period 1998-2018. The authors showed that the interannual variations were typically
small (<3 % per year) and found no evidence of a significant long-term trend in OH over the study
period. Changes in NMHC mole fractions at GEOSummit are well outside what could be explained
by a 3% change in OH tropospheric concentrations. There is, however, likely a difference between
global and regional OH variations (Brenninkmeijer et al., 1992; Spivakovsky et al., 2000;
Lelieveld et al., 2004). In the absence of data on the Arctic and mid-latitudes OH abundance, we
concede that OH may play a role on the observed pause but do not discuss that hypothesis further.
The latter two hypotheses are investigated and verified or rejected in the following sections.
**3.3 No evidence for a change in transport from source regions**
The synoptic-scale tropospheric circulation in the Arctic is driven by three major semi-permanent
pressure systems: i) the Aleutian Low, low-pressure center located south of the Bering Sea area,
ii) the Icelandic Low, low-pressure system located southeast of Greenland near Iceland, and iii)
the Siberian High, high-pressure center located over eastern Siberia (Barrie et al., 1992). During
positive phases of the North Atlantic Oscillation (NAO), the Icelandic Low is strengthened and
transport into the Arctic enhanced, resulting in higher Arctic pollution levels (Duncan and Bey,
2004; Eckhardt et al., 2003). Negative phases of the NAO are associated with decreased transport
from Europe and Siberia and increased transport from North America. In addition, mid-latitude
atmospheric blocking events – quasi-stationary features characterized by a high-pressure cell
centered around 60°N and lasting up to ~15 days (Rex, 1950) – are known to enhance transport of
polluted air to the Arctic (Iversen and Joranger, 1985). Here, we test the hypothesis of a pause in
the growth of atmospheric ethane and propane at GEOSummit driven by the interannual variability
of pollution transport from source regions.
The interannual variability in the origin of air masses influencing GEOSummit was investigated
using April 2012-June 2019 air-mass back trajectories generated with the HYSPLIT model. Figure
5 shows the annual gridded back trajectory frequencies and Figure 6a summarizes the relative
contribution of each geographical sector for each year. Contrary to other Arctic sites (Hirdman et





al., 2010), GEOSummit is mostly influenced by transport from North America and Europe, whereas Siberia has relatively little influence (0-2 %). European air masses represented 3-6 % of the total, with a 10 % high in 2018. The relative contribution of North Atlantic air masses ("ocean") ranged from 1 to 9 %, with a 14 % high from January to August 2019. The frequency of North American air masses exhibited the most variability, ranging from 2 to 20 %. Assuming that the ethane and propane trends are driven by emissions in North America (Helmig et al., 2016) and that these emissions are constant, one would expect higher ethane and propane mixing ratios in years when the relative influence of North American air masses peaked. There is, however, an anticorrelation: a 2-3 % relative contribution of North American air masses in 2014 and 2015 when ethane/propane mixing ratios reached a maximum, and 19 % in 2018 when mixing ratios reached a minimum. This leaves two possibilities: either North American emissions dropped over the studied time period (see Section 3.4), or ethane/propane trends observed at GEOSummit are not driven by emissions in North America (see below).

Local/regional air masses (*i.e.,* around Greenland, see Fig. 5) were the most frequently impacting the site (located near the receptor site). Interestingly, their relative contribution increased from 79 % in 2012 to 91-93 % in 2014-2015 before gradually dropping to 61 % in 2018. The apparent correlation between the relative contribution of local/regional air masses and the ethane/propane trend raises the question of whether these are connected. In order to identify potential sources in this sector, we performed a PSCF analysis to investigate source-receptor relationships (e.g., Pekney et al., 2006; Perrone et al., 2018; Yu et al., 2015; Zhou et al., 2018; Zong et al., 2018). The PSCF calculates the probability that a source is located at latitude $i$ and longitude $j$ (Pekney et al., 2006). Figure S4 shows the results of the PSCF analysis for ethane and propane residuals and shows no consistent pattern associated with elevated concentrations. In both winter and summer, the probability of an ethane or propane source from this analysis is low (<2 % on average).

The history of petroleum exploration activities on the Greenland continental shelf dates back to the 1970s (Arctic Oil & Gas Development: The Case of Greenland, 2020). More recently, the Greenland's government announced the opening of three new offshore areas for exploration in November 2020 (Greenland Opens Offshore Areas for Drilling, 2020). Despite exploration drilling activities, there has never been any O&NG exploitation of Greenland resources (Arctic Oil & Gas Development: The Case of Greenland, 2020). Building on the above, the assumption of a significant local/regional source can be ruled out, and so can the hypothesis that the pause in the





growth of ethane and propane is driven by local/regional emissions. The last remaining hypothesis
is that this pause is due to a change in emissions from any of the other source sectors, or a
combination of them, or total NH emissions and associated change in baseline NH atmospheric
levels. This hypothesis is tested in the following Section using observations at other baseline sites.
**3.4 Evidence for a hemispheric pattern**
Table 1 summarizes the rate of change and 95 % confidence interval for 2010-2014 and 2015-
2018 at Alert (ALT, Nunavut, Canada), Utqiagvik/Barrow (BRW, Alaska, USA), Cape Kumukahi
(KUM, Hawaii, USA), Park Falls (LEF, Wisconsin, USA), and Mace Head (MHD, Ireland – see
Fig. 1) where discrete samples were collected for the NOAA/GML CCGG and HATS cooperative
networks. A clear reversal in interannual changes for ethane and propane mixing ratios is observed
in 2015 at ALT, BRW, KUM, and LEF. These results support the observed changes at
GEOSummit and indicate a hemispheric pattern, likely due to a change in Northern Hemisphere
emissions, with a turning point around late 2014. Biomass burning and anthropogenic activities
being the main emitters of NMHCs, we hereafter focus the discussion on these two sources.
3.4.1    Biomass burning
Occasional biomass burning plumes were observed at GEOSummit. For example, Fig. 7 shows
the simultaneous increase in CO, ethane, propane, and benzene mixing ratios for a short number
of days in July and August 2019. According to the Whole Atmosphere Community Climate Model
(WACCM; Gettelman et al., 2019) CO forecast simulations, available at
https://www.acom.ucar.edu/waccm/forecast/, these enhancements can be attributed to intense
Siberian wildfires occurring at that time (Bondur et al., 2020). In good agreement with the
WACCM simulations, emission ratios (amount of compound emitted divided by that of a reference
compound) derived from these two plumes for ethane and propane (5.4-5.9 $\times 10^{-3}$ and 1.5-1.6
$\times 10^{-3}$ ppb per ppb of CO, respectively; see Fig. S5) are within the range of values reported for
boreal forest and peat fires (Andreae, 2019).
Despite the observation of occasional plumes at GEOSummit, the question remains whether
biomass burning could drive the observed hemispheric pause in the growth of atmospheric ethane
and propane. Figure 6b gives annual biomass burning emissions from all open burning north of
45°N according to the Fire INventory from NCAR (FINNv2.2) emission estimates driven by
MODIS fire detections (Wiedinmyer et al., in prep). Emissions peaked in 2012, known for being
an exceptional wildfire season in the contiguous U.S. (e.g., Lassman et al., 2017; Val Martin et al.,





2013). While propane emissions remained fairly stable from 2014 to 2018, ethane emissions
slightly decreased from 2014 to 2016. However, we did not find any correlation between annual
biomass burning emissions and annually-averaged mixing ratios (true using either 2009-2018 or
2015-2018 data). The seasonal analysis of the correlation between ambient air mixing ratios and
biomass burning emissions yielded similar results. This suggests that the observed pause in the
growth of atmospheric ethane and propane is likely not driven by biomass burning emissions.
This conclusion is further supported by measurements during the global-scale aircraft mission
ATom. Using ethane and propane data collected in the Northern Hemisphere (>20°N) remote free
troposphere during the four ATom seasonal deployments, we found a significant positive
correlation of ethane and propane with tetrachloroethylene ($R^2 = 0.6$, p-value < 0.001) and a poor
correlation with hydrogen cyanide ($R^2 < 0.1$, p-value < 0.001; see Fig. S6), used as tracers of
anthropogenic and biomass burning emissions, respectively (Bourgeois et al., in prep.). These
results from the remote free troposphere confirm that atmospheric ethane and propane ambient air
levels are mostly driven by anthropogenic activities rather than by biomass burning emissions, in
line with results from other studies (e.g., Xiao et al., 2008).

### 3.4.2    O&NG activities

Discrete samples collected at northern-hemisphere baseline sites show that the strongest change
was observed at LEF, located downwind from the Bakken oil field in North Dakota (Gvakharia et
al., 2017), with an increase of ethane mixing ratios of +167.7 [+157.5, +186.0] ppt per year in
2010-2014 and a decrease of -247.8 [-312.2, -158.2] ppt per year in 2015-2018 (see Table 1). This
result, along with previous findings by Helmig et al. (2016) and Franco et al. (2015), supports the
hypothesis that U.S. O&NG emissions could play a major role in driving atmospheric ethane and
propane concentrations in the NH. Here we further discuss this potential contribution to the
observed hemispheric pause in the growth of atmospheric ethane and propane in 2015-2018.
The U.S. has experienced dramatic increases in O&NG production since 2005, underpinned by
technological developments such as horizontal drilling and hydraulic fracturing (Caporin and
Fontini, 2017; Feng et al., 2019). This shale revolution has transformed the U.S. into the world's
top O&NG producer (Gong, 2020). Coincident with the shale gas boom, the U.S. production of
natural gas liquids (ethane, propane, butane, iso-butane, and pentane) has significantly increased
in the past decade from 0.6-0.7 billion barrels in the 2000s to 1.1 billion barrels in 2014, and close
to 1.8 billion barrels in 2019 (U.S. Field Production of Natural Gas Liquids, 2021). Ethane and



propane emissions are primarily due to leakage during the production, processing, and
transportation of natural gas (Tzompa-Sosa et al., 2019; Pétron et al., 2012).
Propane is extracted from natural gas stream and used as a heating fuel. As shown in Figure 8, the
U.S. propane field production temporarily plateaued from June 2014 to December 2016 (U.S. Field
Production of Propane, 2021) due to a slowdown in natural gas production in response to low
natural gas prices. As we consider recent changes in emissions, however, changes in emissions per
unit of production must also be considered. A recent study in the Northeastern Colorado Denver-
Julesburg Basin showed little change in atmospheric hydrocarbons, including propane, in 2008-
2016 despite a 7-fold increase in oil production and nearly tripling of natural gas production,
suggesting a significant decrease in leak and/or venting rate per unit of production (Oltmans et al.,
in review). While we cannot reliably estimate how propane emissions might have changed during
this recent period, these two influences, combined together, could explain the observed temporary
pause in the growth of atmospheric propane.
Estimating the total production, and ultimately emissions, of ethane is even more complex as it
depends on the ethane-to-natural gas price differential. Ethane has long been considered an
unwanted byproduct of O&NG drilling, much of it burned away in the natural gas stream or flared
off at well sites. Today, ethane is a key feedstock for petrochemical manufacturing and the U.S. is
currently the top producer and exporter of ethane (Sicotte, 2020). Depending on the price of ethane
relative to natural gas, ethane can be left in the natural gas stream and sold along with natural gas
– a process known as ethane rejection, or separated at natural gas processing plants along with
other natural gas liquids (such as propane). Assuming the same leak rates for ethane as for methane,
85 % of ethane emissions are due to natural gas extraction and processing, while processed natural
gas transportation and use only represent 15 % of the natural gas supply chain ethane loss rate
(Alvarez et al., 2018). The slowdown in natural gas production from June 2014 to December 2016
(see above) may thus have contributed to the atmospheric ethane plateauing. However, these
estimates do not take into account emissions of ethane from its own supply chain (e.g., separation,
storage, liquefaction for export, ethane cracker to produce ethylene and plastic resins) – for which
leak rates remain unknown. A number of top-down studies, focusing on specific regions or time-
periods (e.g., 2010-2014), have shown that current inventories underestimate ethane emissions
(e.g., Tzompa-Sosa et al., 2017; Pétron et al., 2014). The modeling study led by Dalsøren et al.
(2018) focusing on year 2011 showed that fossil fuel emissions of ethane are likely biased-low by





a factor of 2-3. In this highly dynamic context, where ethane production and volume rejected
continuously vary and where leak rates change over time (Schwietzke et al., 2014), there is a need
for further hemispheric- or global-scale top-down studies focusing on the interannual variability
of ethane emissions.

**4. Summary and Conclusion**
Ethane and propane are the most abundant atmospheric NMHCs and they exert a strong influence
on tropospheric ozone, a major air pollutant and greenhouse gas. Increasing levels have been
reported in the literature from 2009 to 2014, with evidence pointing at U.S. O&NG activities as
the most likely cause (Kort et al., 2016; Helmig et al., 2016; Franco et al., 2016; Hausmann et al.,
2016). The long-term high-resolution records of ambient air $C_2$-$C_7$ NMHCs at GEOSummit
presented here confirm that atmospheric ethane and propane levels increased in the remote arctic
troposphere from 2009 to 2015, but also reveal a pause in their growth in 2015-2018. The analysis
of air-mass back-trajectories allowed us to rule out the possibility that this pause is driven by a
change in transport from source regions. Using independent discrete samples collected at other
NH baseline sites, we show that this pause is observed throughout the northern hemisphere –
suggesting a change in total NH emissions and in baseline NH atmospheric levels. We further
investigated and discussed the contribution of the two main NMHC emitters: biomass burning and
O&NG production. We did not find any correlation between atmospheric ethane and propane
mixing ratios and the FINNv2.2 biomass burning emission estimates. Additionally, data collected
in the NH remote free troposphere during the ATom aircraft campaign support that atmospheric
ethane and propane ambient air levels are mostly driven by anthropogenic activities rather than by
biomass burning emissions. The fact that the strongest rate of change reversal was observed at a
site located downwind from the Bakken oil field in North Dakota tends to suggest that U.S. O&NG
activities yet again played a major role here. The slowdown in U.S. natural gas production from
June 2014 to December 2016 combined with a decrease in leak rate per unit of production could
have contributed to the observed temporary pause. This conclusion is, however, tentative given
the large uncertainties associated with emission estimates, especially with ethane emissions from
its supply chain. We hope this work can be used as a starting point to understand what led to the
pause in the growth of atmospheric ethane and propane in 2015-2018 and, more generally, to what



extent ON&G activities could be responsible for variations in NH baseline ethane and propane
levels.

**Data availability**
All non-methane hydrocarbons and carbon monoxide in-situ data used in this study are archived
and publicly available on the Arctic Data Center database (Angot et al., 2020; Helmig, 2017).
NOAA/GML        HATS        and        CCGG        discrete        data        are        available        at
[ftp://aftp.cmdl.noaa.gov/data/hats/PERSEUS](ftp://aftp.cmdl.noaa.gov/data/hats/PERSEUS) and [ftp://aftp.cmdl.noaa.gov/data/trace_gases/voc/](ftp://aftp.cmdl.noaa.gov/data/trace_gases/voc/),
respectively.

**Author contribution**
DH initiated the long-term monitoring effort at GEOSummit and secured funding over the years.
JH designed and built the GC-FID used for NMHC in-situ monitoring and performed ~bi-annual
on-site visits for maintenance and calibration operations. CD, JC, and BB performed the in-situ
data processing (*i.e.,* GC peak identification, peak integration, background subtraction, and
calculation of mixing ratios). CD, JC, and HA analyzed the data under the supervision of CW and
DH. GP helped evaluating the impact ON&G activities on NMHC trends while IB and CW helped
evaluating the impact of biomass burning. IV, SAM, BRM and JWE provided the NOAA /GML
HATS discrete data. JH and DH provided the NOAA/GML CCGG NMHC discrete data with
contribution from CD, JC, and BB. HA wrote the manuscript with contribution from all co-authors.

**Competing interests**
The authors declare no competing interests.

**Acknowledgements**
We would like to thank the GEOSummit Science Technicians and CH2MHill Polar Services for
their tremendous support in enabling on-site and flask collections at the station. HA, JH, and DH
would like to acknowledge Maria Soledad Pazos, Miguel Orta Sanchez, and all students involved
in the NMHC flask analysis at INSTAAR. IV, SAM, and BRM thank the instrumental analysis
assistance of C. Siso and M. Crotwell and standards prepared and maintained by B. Hall at the



NOAA GML. We would also like to thank Donald Blake, Paul Wennberg, Michelle Kim, Hannah
Allen, John Crounse, and Alex Teng for the ATom dataset used in this analysis.

**Financial support**
The long-term observations and analysis efforts were supported by the National Science
Foundation (grant nos. 1108391 and 1822406) and the NASA ROSES program (grant no.
NNX07AR26G). Undergraduate students Connor Davel and Jashan Chopra received financial
support from the University of Colorado Boulder's Undergraduate Research Opportunities
Program (UROP; grant nos. 7245334 and 5269631, respectively). Support for most CIRES
employees is from NOAA award no. NA17OAR4320101. ATom was funded by NASA ROSES-
2013 NRA NNH13ZDA001N-EVS2.

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





**Table 1:** Rates of change and 95 % confidence interval (in brackets) inferred from discrete sampling (in ppt per year). ALT, BRW, MHD, LEF, and KUM refer to Alert, Utqiagvik/Barrow, Mace Head, Park Falls, and Cape Kumukahi. The localization of the sites can be found in Figure 1. The symbols shown next to each rate of change relate to how statistically significant the estimate is: $p < 0.001$ = ***, $p < 0.01$ = **, and $p < 0.05$ = *.

| Site | 2010-2014 | 2015-2018 |
|------|-----------|-----------|
| Ethane | | |
| ALT | +52.8 [+32.7, +73.0] *** | -56.9 [-79.9, -36.6] *** |
| BRW | +40.5 [+25.9, +59.1] *** | -50.6 [-69.4, -27.6] *** |
| KUM | +18.4 [+7.9, +29.5] *** | -43.1 [-62.1, -28.1] *** |
| LEF | +167.7 [+157.5, +186.0] *** | -247.8 [-312.2, -158.2] *** |
| MHD | +51.8 [+44.4, +63.2] *** | -18.6 [-102.6, +45.4] |
| Propane | | |
| ALT | +24.8 [+16.5, +37.7] *** | -55.6 [-65.1, -45.9] *** |
| BRW | +14.5 [+9.1, +20.2] *** | -35.1 [-45.3, -25.6] *** |
| KUM | +3.1 [+0.2, +5.9] * | -13.2 [-15.9, -10.7] *** |
| LEF | +89.8 [+68.5, +123.5] *** | -110.0 [-173.6, -75.6] *** |
| MHD | +21.3 [+16.9, +27.1] *** | -24.2 [-56.2, -7.2] ** |





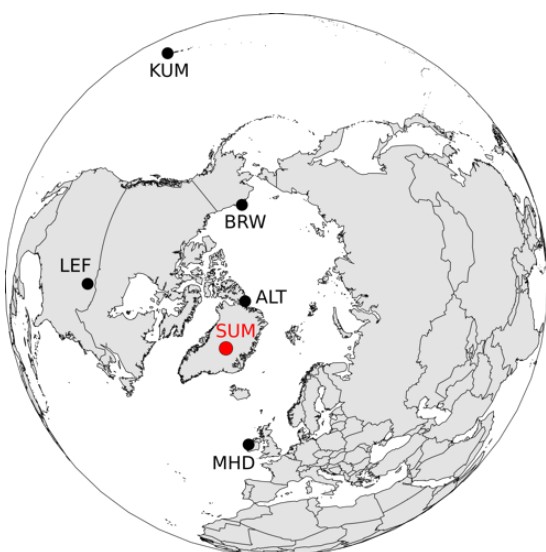

**Figure 1:** Location of the Greenland Environmental Observatory at Summit station (red dot, SUM) where long-term in-situ monitoring was carried out, and of Alert (ALT), Utqiagvik (formerly known as Barrow (BRW)), Mace Head (MHD), Park Falls (LEF), and Cape Kumukahi (KUM), where discrete samples were collected by both the NOAA/ESRL/GML CCGG and HATS sampling networks. The map is centered over the North Pole.

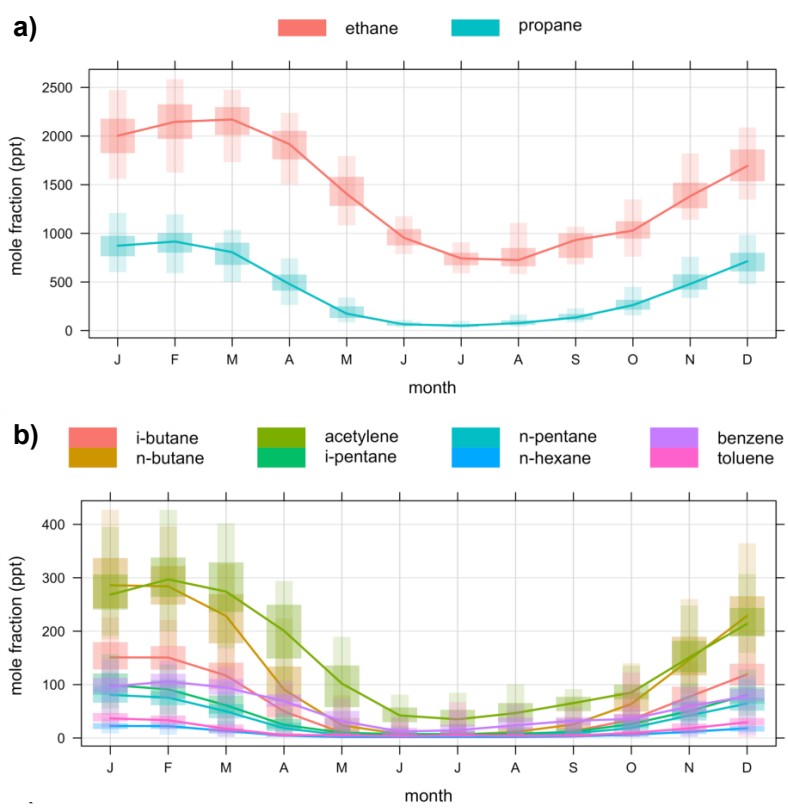

**Figure 2:** Monthly variation of **a)** ethane and propane, and **b)** C$_4$-C$_7$ non-methane hydrocarbons measured in ambient air at GEOSummit as inferred from 2008-2010 and 2012-2020 in-situ measurements. In the monthly boxplots, the lower and upper end of the box correspond to the 25$^{th}$ and 75$^{th}$ percentiles while the whiskers extend from the 5$^{th}$ to the 95$^{th}$ percentiles.



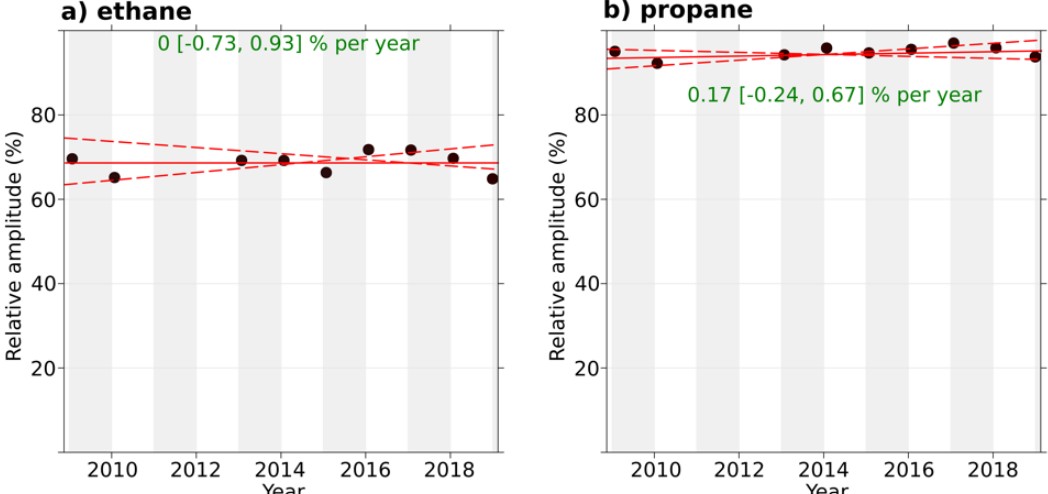

**Figure 3:** Trend in peak-to-minimum seasonal amplitude of **a)** ethane and **b)** propane at GEOSummit, calculated as the relative difference between the maximum and minimum values from the smooth curve for each annual cycle. The solid red line shows the trend estimate and the dashed red lines show the 95 % confidence interval for the trend based on resampling methods. The overall trend is shown at the top along with the 95 % confidence interval in the slope.



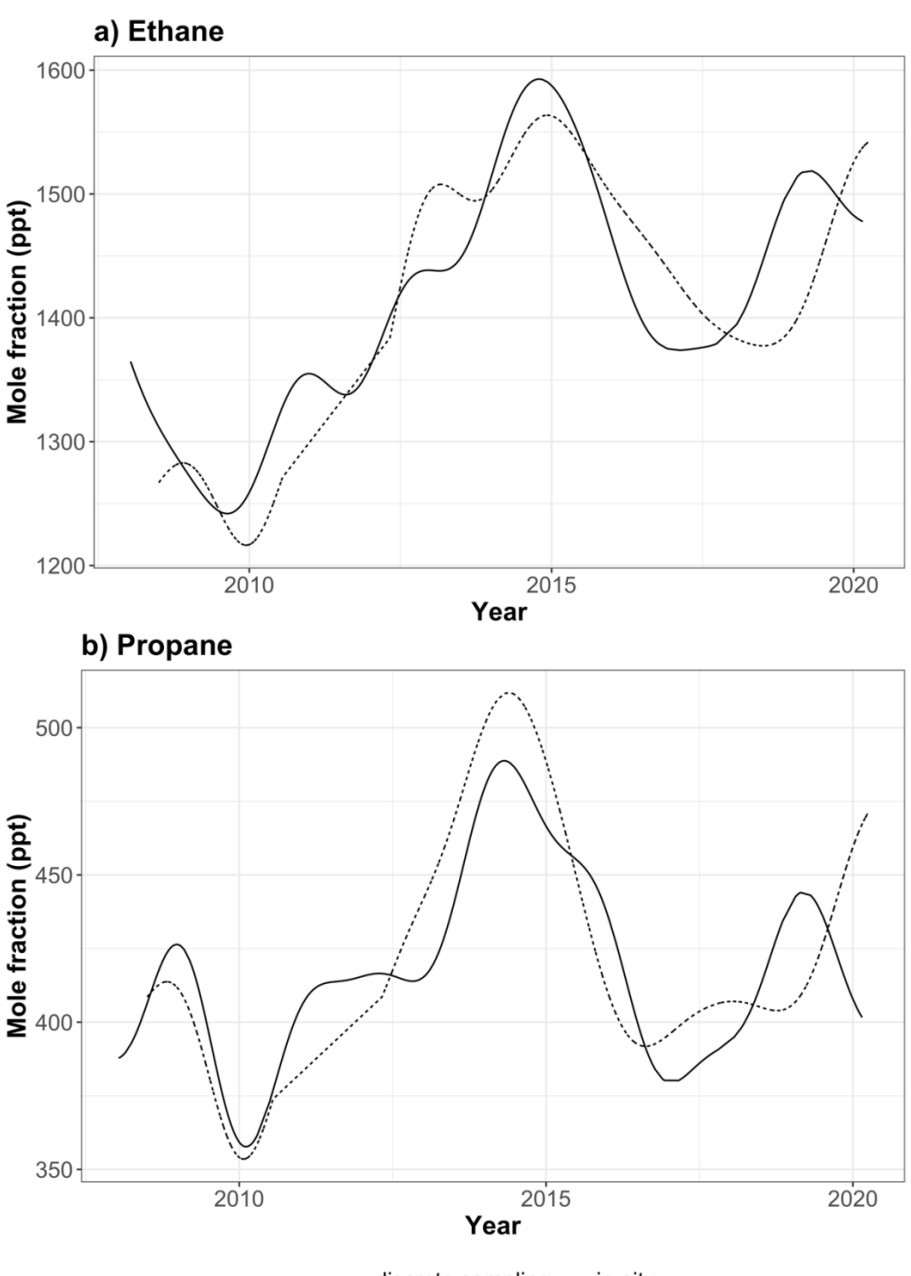

**Figure 4: a)** Ethane, and **b)** propane trends at GEOSummit from July 2008 to March 2020. Trends inferred from in-situ and discrete sampling are shown by the dotted and solid lines, respectively.

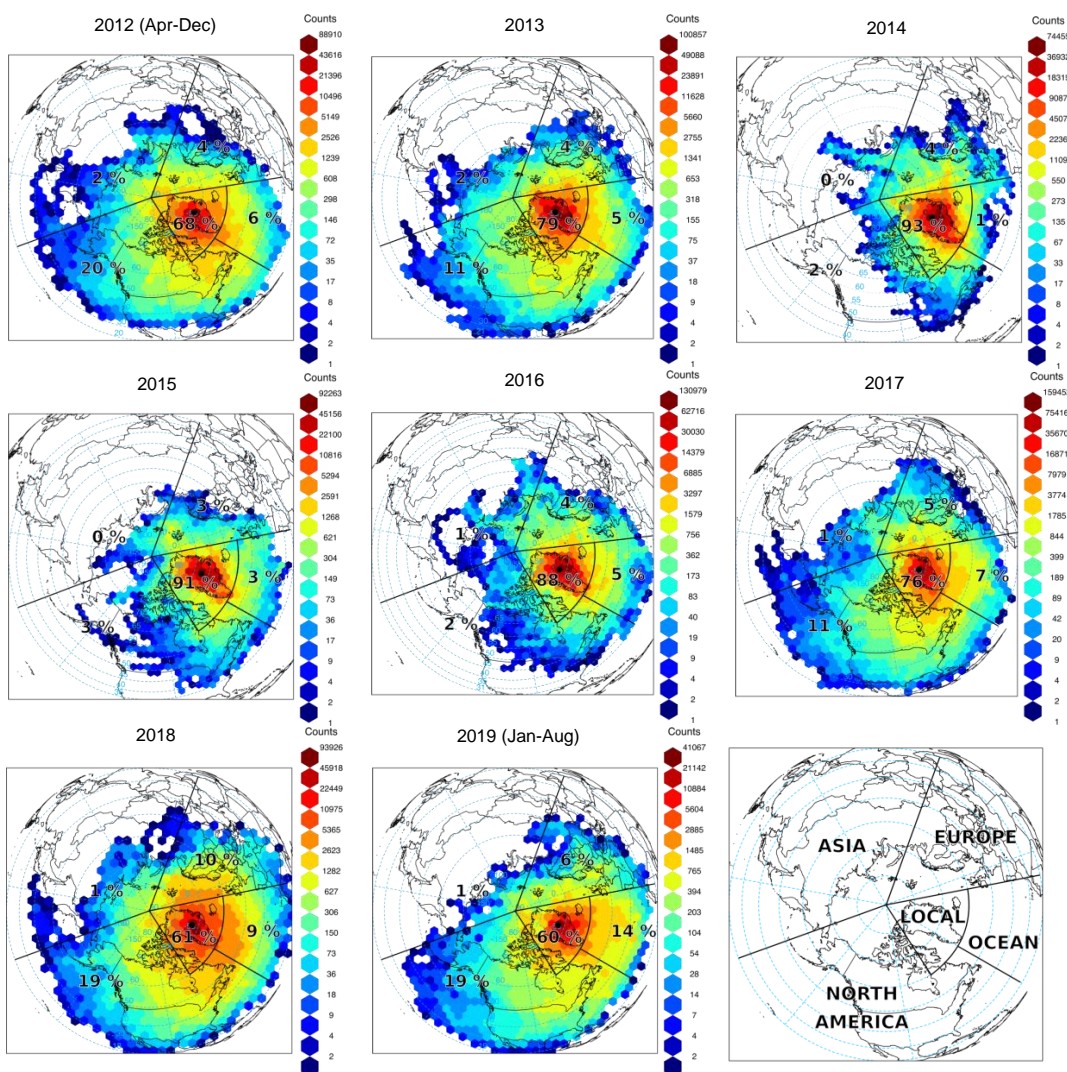

**Figure 5:** Origin air masses influencing GEOSummit (black dot). Gridded back trajectory frequencies using an orthogonal map projection (centered over the North Pole) with hexagonal binning. The tiles represent the number of incidences and the numbers the relative influence of the various sectors.

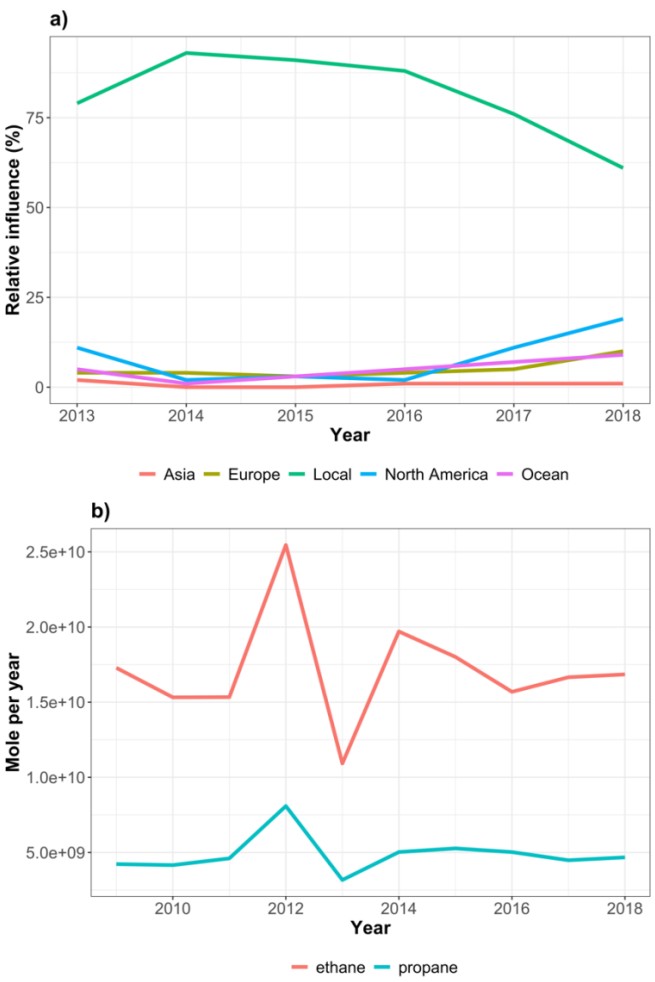

**Figure 6: a)** Annual relative contribution of different geographical sectors to air masses influencing GEOSummit according to the HYSPLIT back-trajectories analysis. **b)** Annual biomass burning emissions (in mole/year) from all open burning north of 45°N according to the Fire INventory from NCAR (FINNv2.2) emission estimates (MODIS only).

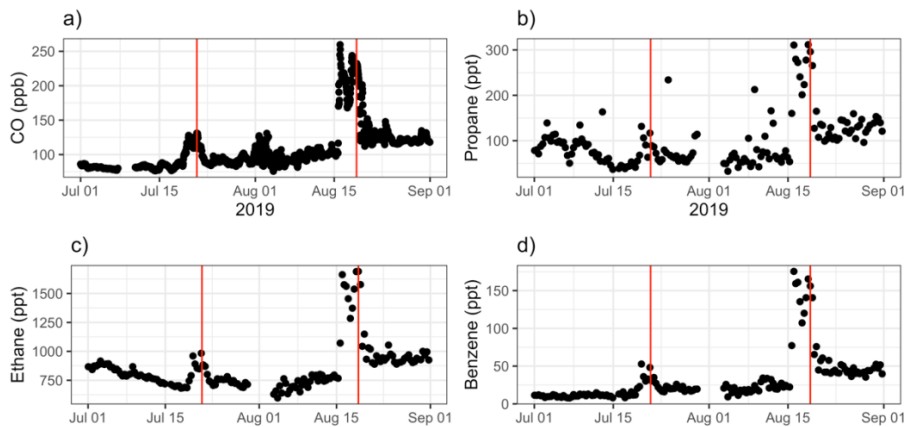

**Figure 7:** Time-series of **a)** carbon monoxide (CO), **b)** propane, **c)** ethane, and **d)** benzene mixing ratios in ambient air at GEOSummit in July-August 2019. The two vertical red lines show the simultaneous enhancement of mixing ratios in two biomass burning plumes.



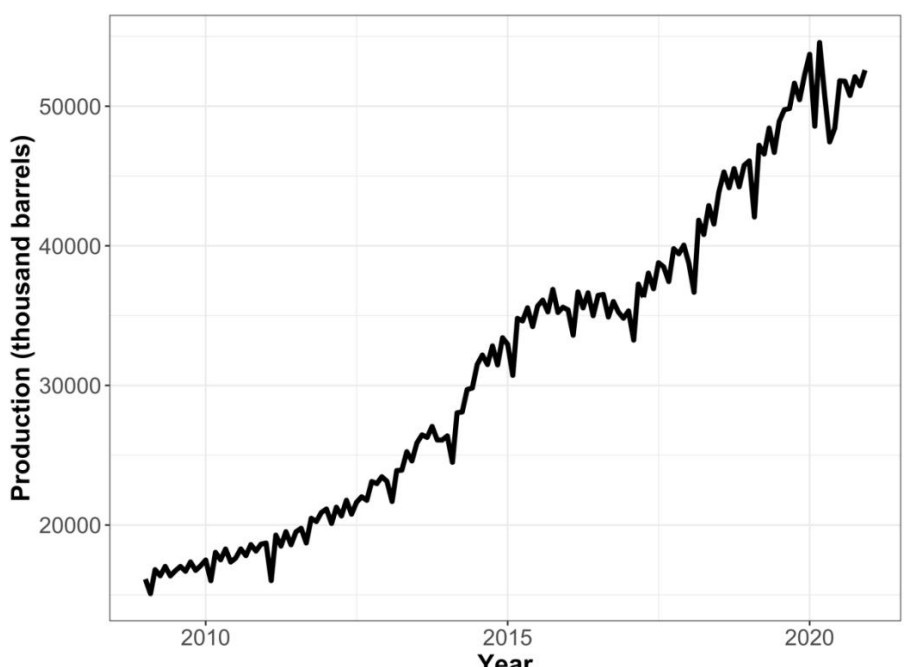

**Figure 8:** U.S. field production of propane in thousand barrels per month. Data courtesy of the U.S. Energy Information Administration. The production temporarily plateaued from June 2014 to December 2016.