# Peer review of "propane in 2015-2018"

_Atmospheric Chemistry and Physics, 2021_

## Author Response (AR2)

Dear Editor,

Please find below (in blue) our responses to your additional comments.

Hélène Angot, on behalf of the authors.
* * *
EC: Editor Comment
RC: Reviewer Comment
AC: Authors Comment

EC: I have gone through your answers to the reviewer questions. In general, I would like you to indicate more clearly how you have addressed the concerns. E.g. if you state that "this has been clarified in the revised manuscript" (e.g. answer to rev. #1 question on transport changes) please explain how it has been clarified. This is usually done by showing and explaining the relevant changes in the manuscript. In addition, some issues remain, which I would like you to clarify.

AC: Our apologies for this lack of clarity. Changes made to the revised manuscript have been more clearly identified and explained (see below, in blue).

EC: Answer to rev #1 about the use of NAO or NAM: I think that this question is not answered. Please discuss the possible use of NAM instead of NAO.

AC: We have edited our answer to reviewer #1 and the manuscript accordingly. Please see edits below (in blue).

EC: line 376 (rev. manuscript): please substantiate this statement with a reference.

AC: Done. This sentence now reads:

"Years with enhanced transport from North America (e.g., 2012, 2019) coincided with a negative NAO index, known to drive decreased (increased) relative contribution from Europe/Asia (North America) (Octaviani et al., 2015)."

EC: Rev. #1: question regarding ENSO effects: your answer states that this effect should be included in the fire statistics you use. But it does not answer the question if there is a fire-modulated relation between the observations and ENSO.

AC: Indeed. We have edited our answer to reviewer #1 accordingly. Please see edits below (in blue).

EC: Rev. #1 and #2 question about 5-day backward trajectories: please explain in the manuscript the reason for using the short 5-day backward trajectories. Computing time for trajectories is actually not very large and does not seem a good reason to only use 5-day trajectories.

AC: The main reason is that results based on 5-day trajectories shown here (GEOSummit mostly influenced by transport from North America and Europe) are in agreement with the isobaric 10-day back-trajectory study by Kahl et al. (1997) and the 20-day backward FLEXPART simulations by Hirdman et al. (2010b). This is mentioned lines 348-349 of the revised manuscript.

EC: rev #2 question about missing references: please show where references have been added.

AC: Done – see below (in blue).

**Response to Murat Aydin (Reviewer 1)**

RC: The paper by Angot et al. analyzes data from the GEOSummit station since 2008. They present data for C2-C7 NMHCs but the analysis primarily focuses understanding the causes of interannual trends in ethane and propane measurements. The paper concludes that the trends are driven primarily by emissions from O&NG industry in North America. The paper is well written, easy to understand, and presentation quality is good. The measurements are based on established methods and traceable calibrations. The analysis is also quite detailed; the authors put in considerable effort to address the different complexities that go into interpretation of short-lived gas measurements from a remote site. The paper will be a valuable contribution to ACP after revisions. My primary concerns are with regards to how possible contributions from transport and biomass burning to the observed interannual trends is addressed (see below). I also listed specific line-by-line comments in the order that they appear in the manuscript.

AC: Thank you for the overall positive feedback. Our responses to the specific comments are provided below.

**Transport (section 3.3)**

RC: Section 3.3 starts out with a brief description of pressure systems that control atmospheric transport and the NAO. NAO is commonly recognized as a decadal oscillation, although the index can go through more rapid phase changes. I'm assuming the observed interannual variability patterns do not correlate with NAO phases? How about Northern Annual Mode, which tends to vary more on interannual time scales?

AC: Thank you for this suggestion. The following sentences have been added to section 3.3:

"We investigated the potential influence of the NAO using monthly mean values from the NOAA Climate Prediction Center. We found a somewhat weak but significant positive correlation between the NAO and monthly-averaged mixing ratios over the 2008-2019 period ($R^2 = 0.4$, p-value $< 0.01$ for both ethane and propane), in line with enhanced transport of pollution to the Arctic during positive phases of the NAO. We also investigated the potential influence of the Northern Annular Mode (NAM), which has a strong interannual component (Hu and Feng, 2010). We found a low correlation between the NAM and monthly-averaged mixing ratios ($R^2 < 0.2$, p-value $= 0.1$ for both ethane and propane). Previous studies have shown that the influence of the NAM varies by regional section of the Arctic; while persistent organic pollutants concentrations were found to correlate with NAM phases at Ny-Ålesund (Svalbard), no correlation was found at Alert (Nunavut, Canada) (Becker et al., 2008; Octaviani et al., 2015)."

RC: The section transitions into the back trajectory analysis in the second paragraph and I struggled to draw a connection between the background provided in the first paragraph onto the second paragraph. I'm not sure how to interpret a back trajectory analysis for investigating the transport variability question for ethane and propane. How far back do the back trajectories go? Mean annual lifetime of ethane is 2 months. In the winter, even the shorter-lived propane can be transported from several weeks away. I find it difficult to dismiss transport changes playing a role in observed interannual trends over Greenland without analysis of data from other regions in the NH. This is done in the following section 3.4 with results from other stations summarized in Table 1. Instead of conclusively rejecting transport contributions in section 3.3, this should be done in conjunction with a more NH wide analysis. Within this context, it would strengthen the paper to show the data that underlie the results shown in Table 1.

AC: The message has been clarified in the revised manuscript. First of all, we no longer state that changes in transport do not play a role here: the title of section 3.3 has been revised accordingly (now: "Changes in transport from source regions") and the following sentences have been deleted in the abstract and conclusion, respectively:

"The analysis of 2012-2019 air mass back-trajectories shows that this pause in mole fraction increases can neither be attributed to changes in atmospheric transport nor to changes in regional emissions."

"The analysis of air-mass back-trajectories allowed us to rule out the possibility that this pause is driven by a change in transport from source regions."

The key message of this section is that changes in transport must be associated with changes in emissions to explain the observed trends (see lines 355-363 of the revised manuscript). Changes in emissions are then discussed in section 3.4. We also tried to better link results from the back-trajectory analysis to the background provided in the first paragraph (and correlation to NAO is discussed above). For instance, we now mention that "years with enhanced transport from North America (e.g., 2012, 2019) coincided with a negative NAO index", which is in line with the background provided in the first paragraph: "Negative phases of the NAO are associated with decreased transport from Europe and Siberia and an increased relative contribution from North America". Regarding the duration of the back-trajectory: we believe that using 5-day backward trajectories is appropriate to get an idea of the origin of air masses (e.g., North America vs. Europe or Siberia). Indeed, the results we show here (GEOSummit mostly influenced by transport from North America and Europe) are in agreement with the isobaric 10-day back-trajectory study by Kahl et al. (1997) and the 20-day backward FLEXPART simulations by Hirdman et al. (2010a). This is mentioned lines 348-349 of the revised manuscript.

**Biomass burning (section 3.4.1)**

RC: The discussions addressing the biomass burning contribution are purely qualitative and leaves some question marks. I think more caveat is required to better convey the full scope of the complexity of the issue. It is established that fossil-fuel sources are larger than biomass burning emissions in the present-day budgets of NMHCs, but biomass burning can still impact variability, especially on interannual time scales. For example, Simpson et al. (GRL, 2006) suggested that ENSO driven variability in biomass burning emissions accounted for most of the observed interannual changes in NH ethane levels during 1996-2004. Did you check any possible correlation with ENSO?

AC: We agree that biomass burning can impact the interannual variability of observed ambient air ethane and propane mixing ratios, and this is actually why we investigate the correlation between observed mixing ratios and biomass burning emissions in section 3.4.1. This is done using the Fire INventory from NCAR (FINNv2.2) emission estimates driven by _daily_ MODIS fire detections (Wiedinmyer et al., 2011). As such, any ENSO driven variability in fire counts (and thus, in biomass burning emissions) should already be taken into account in this analysis. In addition, we did not find any significant correlation between the bi-monthly multivariate ENSO index (MEI.v2; available at https://www.psl.noaa.gov/enso/mei/) and the bi-monthly averaged mixing ratios over the 2008-2019 period ($R^2 = 0.06$, p-value = 0.54 for ethane; $R^2 = 0.01$, p-value = 0.88 for propane).

RC: Correlation analysis will reveal whether a particular source is the primary driver of observed variability, and the lack of correlation between boreal fires and observed gas mixing ratios makes a strong case that there were large changes in ONG emissions during the study period. However, this does not preclude additional significant impacts from biomass burning. Fig. 6b shows max year-to-year changes on the order of 60-70% (0.3-0.5 Tg/y) of total boreal fire emissions. This is equivalent to 50-100 ppt change for ethane over Greenland based on published density estimates (Nicewonger et al., 2020). The paper also only considers boreal fires. It is true that levels of short-lived gases at Summit are much more sensitive to boreal emissions than from low latitude fires, but emission magnitudes also

matter. For ethane, the sensitivity to emissions from boreal fires (roughly 10x the sensitivity from non-boreal emissions) is almost entirely balanced by the larger magnitude of emissions from non-boreal fires (~9x more than boreal) (Nicewonger et al., JGR, 2020). So, if there are correlated changes in boreal and non-boreal fires that are similar strengths in a relative sense (e.g., 50% of each), the impact in ppts could easily reach 100-200 ppt/y level for ethane. Propane is shorter lived so the fire component over Greenland should be dominated by emissions from boreal fires. Emissions from non-boreal fires is another mechanism – in addition to differences in the nature of ONG sources – that can cause Greenland records of ethane and propane to trend differently. The paper should need some justification as to why only boreal fire emissions are considered and why no attempt is made to quantify what the expected contributions are from interannual fire emission variability. What impact does this have over the discussion at the very end of the paper relating propane trends over Greenland to propane production trends shown in Fig. 8?

AC: Thank you for raising this point. In light of the Nicewonger et al. (2020) paper, we agree that only considering boreal fires is a shortcoming. However, we did not find any correlation between observed mixing ratios and Northern Hemisphere (NH) biomass burning emission estimates. We have modified the following paragraph in the revised manuscript and added NH emission estimates to Fig. 6b:

"For ethane, the sensitivity to biomass burning emissions from boreal fires is almost entirely balanced by the larger magnitude of emissions from non-boreal fires (Nicewonger et al., 2020). For propane, being shorter-lived, the fire component over Greenland should be dominated by emissions from boreal fires. We thus investigated the interannual variability of biomass burning emissions from both all open biomass burning north of 45°N (boreal fires) and north of the equator (all NH fires). (…) NH ethane and propane emissions slightly decreased in 2017 and 2018 but remained fairly stable over the 2009-2016 time period. We did not find any significant correlation between annual biomass burning emissions and annually-averaged mixing ratios (true using either 2009-2018 or 2015-2018 data, and true using either all open burning north of 45°N or north of the equator)".

[Figure]

**Revised Figure 6: b)** Annual biomass burning emissions (in mole/year) from all open burning north of 45°N and north of the equator (Northern Hemisphere, NH) according to the Fire INventory from NCAR (FINNv2.2) emission estimates (MODIS only).

**Specific comments**

RC: Line 31: What is meant by regional, Greenland or the Arctic?

AC: This sentence has been removed in the revised manuscript.

RC: Line 36: No need for "however".

AC: Done.

RC: Also, asking for better emission inventories is good, but isn't one of the purposes of long-term measurements networks to provide top-down estimates of emissions? Is this possible for ONG emissions from North America and Europe and what needs to be done to get there? The paper can offer some future direction perhaps?

AC: We agree and this is actually mentioned at the end of section 3.4.2: "A number of top-down studies, focusing on specific regions or time-periods (e.g., 2010-2014), have shown that current inventories underestimate ethane emissions (e.g., Tzompa-Sosa et al., 2017; Pétron et al., 2014). The modeling study led by Dalsøren et al. (2018) focusing on year 2011 showed that fossil fuel emissions of ethane are likely biased-low by a factor of 2-3. In this highly dynamic context, where ethane production and volume rejected continuously vary and where leak rates change over time (Schwietzke et al., 2014), there is a need for further hemispheric- or global-scale top-down studies focusing on the interannual variability of ethane emissions".

RC: Line 82-86: Rephrase or break up the sentence to clarify.

AC: Done.

The original sentence:

"These samples are analyzed for $CO_2$, $CH_4$, CO, $H_2$, $N_2O$, and $SF_6$ at GML (e.g., Geller et al., 1997; Komhyr et al., 1985; Steele, 1991), at the University of Colorado Institute for Arctic and Alpine Research (INSTAAR) for stable isotopes of $CO_2$ and $CH_4$ (Miller et al., 2002; Trolier et al., 1996), and, since 2004, for a variety of volatile organic compounds (VOCs) including $C_2$-$C_7$ NMHCs (Pollmann et al., 2008; Schultz et al., 2015)."

now reads:

"These samples are analyzed for $CO_2$, $CH_4$, CO, $H_2$, $N_2O$, and $SF_6$ at GML (e.g., Geller et al., 1997; Komhyr et al., 1985; Steele, 1991), and at the University of Colorado Institute for Arctic and Alpine Research (INSTAAR) for stable isotopes of $CO_2$ and $CH_4$ (Miller et al., 2002; Trolier et al., 1996). These samples have also been analyzed for a variety of volatile organic compounds (VOCs) including $C_2$-$C_7$ NMHCs at INSTAAR since 2004 (Pollmann et al., 2008; Schultz et al., 2015)."

RC: Line 117: Replace "i.e.," with which.

AC: Done.

RC: Line 203: Grouped instead of "filtered out"?

AC: "Filtered out" has been replaced by "removed" in the revised manuscript.

RC: Line 248-250: Is there a significant correlation without ethane in Fig. S1? I'm not sure what inference to draw from this figure; some very short-lived gases have significant local sources during summer and not the others, or measurement noise (blanks?) is significant for some gases when levels are too low?

AC: On second thought, this Figure does not bring anything and has been removed from the revised manuscript.

RC: Line 301-302: Changes in instead of "a change in".

AC: Done.

RC: Line 335-338: How far back do the back trajectory go?

AC: As mentioned in the Methods section, we used 5-day air-mass back trajectories.

RC: Line 368: Possibility of instead of "assumption of".

AC: Done.

RC: Line 370: Is there fire activity in or very near Greenland?

AC: Fires can occur in Greenland but are not frequent.
(https://earthobservatory.nasa.gov/images/145302/another-fire-in-greenland).

RC: Line 375 – Table 1: Are the trends in this table determined from single year averages for end-point years or do they reflect linear fits to de-seasonalized time series data? Showing the data would be preferable, perhaps in the supplement.

AC: The trend analysis was done as described in section 2.4, i.e., using de-seasonalized time-series. The ethane and propane time-series at the different Northern Hemisphere sites have been included in the revised supplement. See Figures S4 and S5 of the revised supplement.

RC: Line 395: Is Fig. S5 all the data visible in Fig. 7, or just the plume? If just the plume, indicate how you define the plume, and it would be interesting to see how the property-property plots for the entire data set from July-Aug 2019 look like.

AC: We assume you actually refer to Fig. S4. As mentioned in the caption, we only used data from the biomass burning plumes. The caption has been revised and now includes the following sentence: "This figure was made using data from July 14-23, 2019 and from August 15-23, 2019 for the July and August biomass burning plumes, respectively". For your reference, please find below the plots for the entire July-Aug 2019 dataset. Emission ratios derived from these two methods (plume vs. entire dataset) are similar – that was a good sanity check though, thank you for asking.

[Figure]

Figure R1: Scatter plot of ethane, propane, and benzene vs. carbon monoxide (CO) mixing ratios observed at GEOSummit in July and August 2019. The red line gives the fitted linear regression with the 95 % confidence interval (grey shaded region). The slope, given at the top, gives the emission ratio (amount of compound emitted divided by that of a reference compound (CO here)).

**Response to Reviewer 2**

RC: In this study, Angot et al. present an analysis of the long-term dataset (2008-2010, 2012-2020) of NMHCs in the arctic site of GEOSummit. Their findings show that the observed increasing trend of ethane and propane from mid 2009 to mid 2014 reversed from 2015-2018 temporarily. They found the decreasing trend likely due to a slowdown in U.S. natural gas production and a decrease in the leaking rate per unit of production. The paper is generally well written and is detailed when presenting data, findings, plausible explanations, and conclusions. This paper contributes to the scientific understanding of the impact of oil and gas emissions on atmospheric trace gases. Moreover, observations in the arctic regions are particularly important for models, which tend to misrepresent polar regions. I recommend this paper for publication after minor revisions.

AC: Thank you for the overall positive feedback. Our responses to the specific comments are provided below.

RC: My biggest concern is how section 3.3 is presented. I found the whole section confusing to read. First, the title says there is no evidence for change in transport from source regions, but the HYSPLIT analysis and the same section mentions there are important interannual changes in the transport from source regions. Also, I was surprised to see HYSPLIT results show that the site was mostly impacted by local/regional air masses. This made me wonder if the decision of a 5-day backward trajectory should be revised and increased in order to capture the transport from source regions as the title suggest.

AC: The message has been clarified in the revised manuscript. First of all, we no longer state that changes in transport do not play a role here: the title of section 3.3 has been revised accordingly (now: "Changes in transport from source regions") and the following sentences have been deleted in the abstract and conclusion, respectively:

"The analysis of 2012-2019 air mass back-trajectories shows that this pause in mole fraction increases can neither be attributed to changes in atmospheric transport nor to changes in regional emissions."

"The analysis of air-mass back-trajectories allowed us to rule out the possibility that this pause is driven by a change in transport from source regions."

The key message of this section is that changes in transport must be associated with changes in emissions to explain the observed trends (see lines 355-363 of the revised manuscript). Changes in emissions are then discussed in section 3.4. We also tried to better link results from the back-trajectory analysis to the background provided in the first paragraph (see comments by reviewer 1). Regarding the duration of the back-trajectories: we believe that using 5-day backward trajectories is appropriate to get an idea of the origin of air masses (e.g., North America vs. Europe or Siberia). Indeed, the results we show here (GEOSummit mostly influenced by transport from North America and Europe) are in agreement with the isobaric 10-day back-trajectory study by Kahl et al. (1997) and the 20-day backward FLEXPART simulations by Hirdman et al. (2010). This is now mentioned lines 348-349 of the revised manuscript. Considering the computing time required to generate the trajectories and the fact that we obtain results in good agreement with the literature, we believe generating longer trajectories would not bring anything new to the study.

**Specific comments**

RC: The authors miss to provide references in various sentences. Sometimes it is unclear whether the results presented correspond to this study or a previous one. I marked the most important sentences where references are missing and suggest doing a thorough revision of the paper by the authors to correct this.

AC: Thank you for pointing that out. The manuscript has been carefully revised to include missing references. See specific comments below.

RC: Change wording of Lines 429-431 because it is almost copied word by word from the first line in section 3.1.1 in Tzompa-Sosa et al., 2019. Also, I suggest adding Roest and Schade (2017) as a reference.

AC: Done.

This sentence now reads:

"The main source of ethane and propane has been identified to be leakage during the production, processing, and transportation of natural gas (Tzompa-Sosa et al., 2019; Pétron et al., 2012; Roest and Schade, 2017)".

Instead of:

"Ethane and propane emissions are primarily due to leakage during the production, processing, and transportation of natural gas (Tzompa-Sosa et al., 2019; Pétron et al., 2012)".

RC: Lines 277-279. Reference needed in this sentence.

AC: Done.

"As a consequence, atmospheric ethane background air mixing ratios significantly declined during 1984-2010, by an average of -12.4 ± 1.3 ppt per year in the Northern Hemisphere (Aydin et al., 2011; Worton et al., 2012; Helmig et al., 2014)".

RC: Lines 279-282. It is unclear these results correspond to the present study or to a previous one. If If the latter, reference is needed.

AC: This has been clarified in the revised manuscript.

However, the analysis by Helmig et al. (2016) of ten years (2004-2014) of NMHC data from air samples collected at NOAA GML remote global sampling sites (including GEOSummit) showed (…)".

RC: Lines 405-409. There is no reference to the time frame and sampling locations/ares of ATOM observations considered here. A detailed explanation of the data considered is needed.

AC: This has been clarified in the revised manuscript: "This conclusion is further supported by measurements during the aircraft mission ATom over the Pacific and Atlantic Oceans. Using ethane and propane data collected in the Northern Hemisphere (>20°N) remote free troposphere during the four ATom seasonal deployments (July-August 2016, January-February 2017, September-October 2018, and April-May 2018), we found …".

**Technical corrections**

RC: Line 289. Suggest changing "on the year 2015 reversal" to "on the 2015-2018 reversal period".

AC: Done.

RC: Lines 293-294. Suggest adding "(dotted lines)" to this sentence, because the solid line is the predominant line, it tends to be the one the reader focuses on.

AC: Done. Good point, thank you for the suggestion!

This sentence now reads:

"Figure 4a shows the July 2008-March 2020 ethane trend at GEOSummit, as inferred from our in-situ measurements (dotted line)".

**References**

Becker, S., Halsall, C. J., Tych, W., Kallenborn, R., Su, Y., and Hung, H.: Long-term trends in atmospheric concentrations of α- and γ-HCH in the Arctic provide insight into the effects of legislation and climatic fluctuations on contaminant levels, Atmos. Environ., 42, 8225–8233, https://doi.org/10.1016/j.atmosenv.2008.07.058, 2008.

Dalsøren, S. B., Myhre, G., Hodnebrog, Ø., Myhre, C. L., Stohl, A., Pisso, I., Schwietzke, S., Höglund-Isaksson, L., Helmig, D., Reimann, S., Sauvage, S., Schmidbauer, N., Read, K. A., Carpenter, L. J., Lewis, A. C., Punjabi, S., and Wallasch, M.: Discrepancy between simulated and observed ethane and propane levels explained by underestimated fossil emissions, Nat. Geosci., 11, 178–184, https://doi.org/10.1038/s41561-018-0073-0, 2018.

Geller, L. S., Elkins, J. W., Lobert, J. M., Clarke, A. D., Hurst, D. F., Butler, J. H., and Myers, R. C.: Tropospheric SF6: Observed latitudinal distribution and trends, derived emissions and interhemispheric exchange time, Geophys. Res. Lett., 24, 675–678, https://doi.org/10.1029/97GL00523, 1997.

Hirdman, D., Burkhart, J. F., Sodemann, H., Eckhardt, S., Jefferson, A., Quinn, P. K., Sharma, S., Ström, J., and Stohl, A.: Long-term trends of black carbon and sulphate aerosol in the Arctic: changes in atmospheric transport and source region emissions, Atmos Chem Phys, 10, 9351–9368, https://doi.org/10.5194/acp-10-9351-2010, 2010a.

Hirdman, D., Sodemann, H., Eckhardt, S., Burkhart, J. F., Jefferson, A., Mefford, T., Quinn, P. K., Sharma, S., Ström, J., and Stohl, A.: Source identification of short-lived air pollutants in the Arctic using statistical analysis of measurement data and particle dispersion model output, Atmospheric Chem. Phys., 10, 669–693, https://doi.org/10.5194/acp-10-669-2010, 2010b.

Hu, Q. and Feng, S.: Influence of the Arctic oscillation on central United States summer rainfall, J. Geophys. Res. Atmospheres, 115, https://doi.org/10.1029/2009JD011805, 2010.

Kahl, J. D. W., Martinez, D. A., Kuhns, H., Davidson, C. I., Jaffrezo, J.-L., and Harris, J. M.: Air mass trajectories to Summit, Greenland: A 44-year climatology and some episodic events, J. Geophys. Res. Oceans, 102, 26861–26875, https://doi.org/10.1029/97JC00296, 1997.

Komhyr, W. D., Gammon, R. H., Harris, T. B., Waterman, L. S., Conway, T. J., Taylor, W. R., and Thoning, K. W.: Global atmospheric CO2 distribution and variations from 1968–1982 NOAA/GMCC CO2 flask sample data, J. Geophys. Res. Atmospheres, 90, 5567–5596, https://doi.org/10.1029/JD090iD03p05567, 1985.

Miller, J. B., Mack, K. A., Dissly, R., White, J. W. C., Dlugokencky, E. J., and Tans, P. P.: Development of analytical methods and measurements of 13C/12C in atmospheric CH4 from the NOAA Climate Monitoring and Diagnostics Laboratory Global Air Sampling Network, J. Geophys. Res. Atmospheres, 107, ACH 11-1-ACH 11-15, https://doi.org/10.1029/2001JD000630, 2002.

Nicewonger, M. R., Aydin, M., Prather, M. J., and Saltzman, E. S.: Extracting a History of Global Fire Emissions for the Past Millennium From Ice Core Records of Acetylene, Ethane, and Methane, J. Geophys. Res. Atmospheres, 125, e2020JD032932, https://doi.org/10.1029/2020JD032932, 2020.

Octaviani, M., Stemmler, I., Lammel, G., and Graf, H. F.: Atmospheric Transport of Persistent Organic Pollutants to and from the Arctic under Present-Day and Future Climate, Environ. Sci. Technol., 49, 3593–3602, https://doi.org/10.1021/es505636g, 2015.

Pétron, G., Karion, A., Sweeney, C., Miller, B. R., Montzka, S. A., Frost, G. J., Trainer, M., Tans, P., Andrews, A., Kofler, J., Helmig, D., Guenther, D., Dlugokencky, E., Lang, P., Newberger, T., Wolter, S., Hall, B., Novelli, P., Brewer, A., Conley, S., Hardesty, M., Banta, R., White, A., Noone, D., Wolfe,

D., and Schnell, R.: A new look at methane and nonmethane hydrocarbon emissions from oil and natural gas operations in the Colorado Denver-Julesburg Basin, J. Geophys. Res. Atmospheres, 119, 6836–6852, https://doi.org/10.1002/2013JD021272, 2014.

Pollmann, J., Helmig, D., Hueber, J., Plass-Dülmer, C., and Tans, P.: Sampling, storage, and analysis of C2–C7 non-methane hydrocarbons from the US National Oceanic and Atmospheric Administration Cooperative Air Sampling Network glass flasks, J. Chromatogr. A, 1188, 75–87, https://doi.org/10.1016/j.chroma.2008.02.059, 2008.

Schultz, M. G., Akimoto, H., Bottenheim, J., Buchmann, B., Galbally, I. E., Gilge, S., Helmig, D., Koide, H., Lewis, A. C., Novelli, P. C., Dülmer, C. P.-, Ryerson, T. B., Steinbacher, M., Steinbrecher, R., Tarasova, O., Tørseth, K., Thouret, V., and Zellweger, C.: The Global Atmosphere Watch reactive gases measurement network, Elem Sci Anth, 3, 000067, https://doi.org/10.12952/journal.elementa.000067, 2015.

Schwietzke, S., Griffin, W. M., Matthews, H. S., and Bruhwiler, L. M. P.: Natural Gas Fugitive Emissions Rates Constrained by Global Atmospheric Methane and Ethane, Environ. Sci. Technol., 48, 7714–7722, https://doi.org/10.1021/es501204c, 2014.

Steele, L. P.: Atmospheric Methane Concentrations, the NOAA/CMDL Global Cooperative Flask Sampling Network, 1983-1988, Oak Ridge National Laboratory, 324 pp., 1991.

Stohl, A.: Computation, accuracy and application of trajectories - a review and bibliography, Atmos. Environ., 32, 947–966, 1998.

Trolier, M., White, J. W. C., Tans, P. P., Masarie, K. A., and Gemery, P. A.: Monitoring the isotopic composition of atmospheric CO2: Measurements from the NOAA Global Air Sampling Network, J. Geophys. Res. Atmospheres, 101, 25897–25916, https://doi.org/10.1029/96JD02363, 1996.

Tzompa-Sosa, Z. A., Mahieu, E., Franco, B., Keller, C. A., Turner, A. J., Helmig, D., Fried, A., Richter, D., Weibring, P., Walega, J., Yacovitch, T. I., Herndon, S. C., Blake, D. R., Hase, F., Hannigan, J. W., Conway, S., Strong, K., Schneider, M., and Fischer, E. V.: Revisiting global fossil fuel and biofuel emissions of ethane, J. Geophys. Res. Atmospheres, 122, 2493–2512, https://doi.org/10.1002/2016JD025767, 2017.

Wiedinmyer, C., Akagi, S. K., Yokelson, R. J., Emmons, L. K., Al-Saadi, J. A., Orlando, J. J., and Soja, A. J.: The Fire INventory from NCAR (FINN): a high resolution global model to estimate the emissions from open burning, Geosci. Model Dev., 4, 625–641, https://doi.org/10.5194/gmd-4-625-2011, 2011.